# FreePIH: Training-Free Painterly Image Harmonization with Diffusion Model

## ABSTRACT

This paper provides an efficient training-free painterly image harmonization (PIH) method, dubbed *FreePIH*, that leverages only a pre-trained diffusion model to achieve state-of-the-art harmonization results. Unlike existing methods that require either training auxiliary networks or fine-tuning a large pre-trained backbone, or both, to harmonize a foreground object with a painterly-style background image, our *FreePIH* tames the denoising process as a plug-in module for foreground image style transfer. Specifically, we find that the very last few steps of the denoising (i.e., generation) process strongly correspond to the stylistic information of images, and based on this, we propose to augment the latent features of both the foreground and background images with Gaussians for a direct denoising-based harmonization. To guarantee the fidelity of the harmonized image, we make use of multi-scale features to enforce the consistency of the content and stability of the foreground objects in the latent space, and meanwhile, aligning both fore-/back-grounds with the same style. Moreover, to accommodate the generation with more structural and textural details, we further integrate text prompts to attend to the latent features, hence improving the generation quality. Quantitative and qualitative evaluations on COCO and LAION 5B datasets demonstrate that our method can surpass representative baselines by large margins.

## CCS CONCEPTS

• **Computing methodologies** → **Computer vision**; Reconstruction.

## KEYWORDS

Diffusion Model, Image Harmonization, Image Editing.

## 1 INTRODUCTION

Image compositing is a fundamental task in image editing, enabling users to merge foreground and background images to create new artwork. Often, the foreground and background images have varying colors and structures. Our objective is to seamlessly integrate a foreground item into a different background image, producing a natural and visually pleasing result. In recent years, there has been a growing interest in image compositing, with researchers exploring various methods to improve the quality of composite images [3, 5, 13, 23]. The overall framework utilized in these studies

*ACM MM, 2024, Melbourne, Australia*

© 2024 Copyright held by the owner/author(s). Publication rights licensed to ACM.
ACM ISBN 978-x-xxxx-xxxx-x/YY/MM
https://doi.org/10.1145/nnnnnnn.nnnnnnn

remains largely unchanged. Typically, a set of loss terms (such as content loss, style loss, and stability loss) is designed based on a pre-trained feature extractor network, often VGG-19 [40]. These loss terms are then iteratively optimized to update the pixel values of the foreground image. One notable advantage of this framework is its avoidance of the need for additional data collection or resource-intensive model training or fine-tuning processes, making it a plug-in module built upon an off-the-shelf pre-trained model.

Recent advancements in image generation techniques have led researchers to explore the use of Text to Image Diffusion Models (T2I-DM) for image compositing [20, 24, 34, 37, 42]. T2I-DM[9, 17, 32, 34], combined with CLIP [35], enables users to generate images based on natural language prompts. However, one of the challenges faced with T2I-DM is the loss of control over the generated images. In scenarios where users want to composite specific items into an image and describe them using natural language, T2I-DM may not generate the desired output or may be difficult to guide using simple words. Previous work, such as Dreambooth [38, 45], can inject specific items into output images, but this approach requires lengthy fine-tuning processes. Other works like text-driven editing approaches[1, 4, 18, 21, 30, 43] are insufficient for image compositing, as it is sometimes challenging to provide accurate verbal representations to capture the details or preserve the identity and appearance of a given object image.

Enabling T2I-DM with image compositing capabilities can be achieved through a direct approach involving the use of T2I-DM to generate the background image, followed by the application of an image compositing algorithm to blend the foreground item into the generated image. However, this method often yields a foreground image that does not harmonize with the style of the background, resulting in unnatural fused results. Additionally, previous approaches typically utilize VGG-19 as the feature extractor, necessitating the use of both the neural network module employed by the T2I-DM and VGG-19 for compositing. However, we find that the modules within T2I-DM already serve as effective multi-scale feature extractors. Unlike VGG-19, which is trained for classification, the modules in T2I-DM are specifically trained for image generation and possess the ability to extract multi-scale feature maps, thereby surpassing VGG-19 in handling the complex feature representations required for image editing.

In this paper, we leverage the pre-trained T2I-DM to conduct image compositing and introduce a method named "FreePIH". The background can either be the images generated by the T2I-DM or provided by the user. This approach differs from text-driven editing with diffusion models, as it allows users to add items to specific locations while preserving their structure and appearance, providing greater control over their AIGC (Artificial Intelligence Generated Content) artworks. Specifically, our framework takes a tuple $(\mathbf{x}_G, \mathbf{x}_I, m)$ that represents the background, foreground, and mask, respectively. We utilize the Variational Autoencoder (VAE) image encoder in T2I-DM to extract low-level content feature maps.

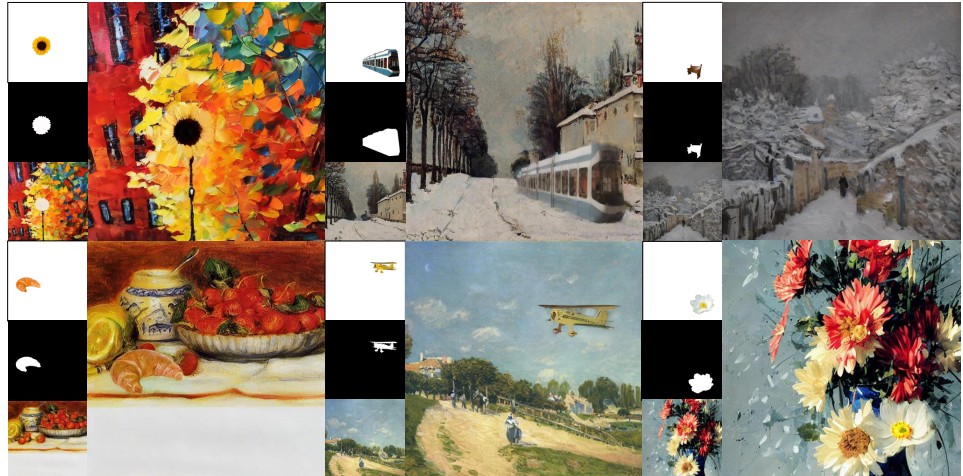

**Figure 1: Example of painterly image harmonization with our proposed FreePIH.**

While methods like ControlNet [48, 51], T2I-adapter[31, 47] can achieve controllable generation, they are only suitable for scenarios where there is no expected output. For instance, if we want a sunflower with specific details in the target position, these methods often fail to preserve the provided details, resulting in a different sunflower. Our work contributes by enabling this type of controllable editing while preserving the details of the provided images.

To achieve harmonious image composition, we design a two-branch fusion process based on the pipeline of DM. Foreground and background images follow different branches during the composition process. For the background branch, after injecting noise, background images follow the normal DM denoising pipeline to remove the added noise step by step. For the foreground branch, we use different features to control the denoising process in order to achieve content preservation and style transfer simultaneously. Specifically, to obtain high-level style feature maps, we first apply data augmentation to the latent features extracted by the VAE based on the diffusion forward process. Based on three key observations: (1) the denoising process in DM can be divided into two parts[8, 14], with the early denoising steps generating the overall structure and outline, while the remaining denoising steps refine the details and style gradually. (2) Injecting excessive noise can disrupt the structure of the original image and lead to a loss of control over the image content. (3) With only a small amount of noise injection, we only need to denoise a few steps, thereby accelerating the overall harmonization process. As a result, to ensure consistency between input and output images and save inference time, the level of forward noise injection in the data augmentation needs to be controlled. The augmented latent features then interact with the text input in the DM module using a cross-attention mechanism. The resulting feature map serves as our high-level style representation. Through iterative optimization, we seamlessly blend the foreground into the background. Importantly, our method eliminates the need for model fine-tuning, distinguishing it from existing baselines, particularly DM-based approaches which rely on auxiliary modules and

hundreds of GPU hours for image fusion training. Furthermore, our method empowers T2I-DM users with increased control over their AIGC artworks.

Our contributions can be summarized as follows:

- We propose FreePIH, which can work as a plug-in module to enable image harmonization on off-the-shelf T2I-DM without the need for collecting new data, training auxiliary networks, and fine-tuning pre-trained models.
- We conduct noise augmentation on the latent features and leverage the corresponding output to accurately capture stylistic information based on the feature of DM.
- With the composition capacity of FreePIH, users gain enhanced autonomy in shaping their AIGC artworks when using DM-based genetative model.
- Qualitative and quantitative analysis reveals that our *FreePIH* method can generate more natural fusion images compared with other baselines.

## 2 RELATED WORK

### 2.1 Image Compositing

Image compositing has historically presented a formidable challenge in the realm of image editing, aiming to seamlessly integrate a given foreground image into a target background image. The prevailing body of work is rooted in the framework devised by [11]. This framework employs a pre-trained neural network model to extract multi-scale feature maps, which are subsequently utilized to compute a set of loss terms encompassing style loss, content loss, and stability loss. Subsequent endeavors have concentrated on refining the design of these loss terms and the feature extractor.

Certain studies, such as DPH[27] and DIB[49], optimize the optimization process by integrating diverse loss terms such as Poisson image loss and histogram loss. Others, including PHDNet[3] and FDIT[2], have ascertained that transforming the feature maps into the frequency domain can enhance appearance preservation

and compositing harmonization. Furthermore, CNN-based networks such as RainNet[23], DoveNet [6], and BAIN [13], as well as attention-based networks like SAM[7] and CDTNet [5], have been enlisted to supplant the former VGG-19 feature extractor. Diverging from the aforementioned research, we harness the latent potential of off-the-shelf T2I-DM models to capitalize on their image harmonization capabilities. Given that the modules within the T2I-DM models are trained on a large-scale dataset for image generation, the features extracted by these modules inherently serve as zero-shot multi-scale feature extractors for image compositing and harmonization tasks.

## 2.2 Text to Image Diffusion

The Diffusion Model (DM)[16] represents a pioneering AI model inspired by non-equilibrium thermodynamics. It functions by establishing a Markov chain of diffusion steps, gradually introducing random noise to data, and subsequently learning to reverse the diffusion process to generate desired data samples from the noise. In the realm of image generation, DM initiates the process by generating a random Gaussian noise image and progressively eliminates noise in a step-by-step fashion until a clear image is obtained.

Through integration with the CLIP text encoder, DM acquires the capability to employ natural language prompts to steer the diffusion generation process. Noteworthy models such as Stable Diffusion[37], DallE-[36], and Midjourney have showcased remarkable proficiency in executing text-to-image (T2I) guided generation. However, while T2I-DM serves as a potent tool for image generation, there are scenarios where the generated images may not align with the user's expectations. For instance, if the desired prompt is "dog sitting in front of a door," providing T2I-DM with the text prompt "dog" might yield an entirely different image with a dog sitting elsewhere. Consequently, the challenge of providing users with enhanced control over their AI-generated artworks persists.

## 2.3 Guide Diffusion

To enhance the control of DM, researchers have proposed several updated versions of T2I-DM. Works such as Dreambooth[38] and text-inversion[10] address this issue by tailoring T2I-DM generation to users' personalized requirements. However, these methods necessitate extensive time and computational resources, as they involve hours of fine-tuning a pre-trained T2I-DM model. Additionally, Dreambooth requires a significant amount of images with similar semantic content, which may not always be readily available.

An alternative approach, known as prompt-to-prompt, facilitates image editing by modifying the cross-attention modules of T2I-DM[15, 44]. However, the edited images generated through this method are limited to those produced by T2I-DM itself, as the attention editing operation relies on the previous attention feature map. Consequently, it is not ideal for editing user-provided images.

Recent research endeavors have attempted to equip T2I-DM with text-driven editing capabilities[1]. Nevertheless, accurately and concisely describing a personalized demand can be challenging in some cases. Furthermore, even when an appropriate text prompt is provided (e.g., a precise description of a dog's ears, eyes, and nose

features), T2I-DM may capture these features but ultimately generate a dog with completely different characteristics. This diverges from the original intention of having the dog appear in the desired background.

## 2.4 Diffusion for Image Harmonization

Over the last year, many efforts have been made to adapt the powerful pre-train DM into the image harmonization tasks. For instance, the CDC [12] introduced a technique for conditioning at the time of inference that incorporates high-frequency background details and low-frequency foreground style for image creation. However, the assumption made by CDC that high-frequency and low-frequency features in an image always represent style and content information respectively is not always accurate. The InST[50] project was inspired by the idea that a one-of-a-kind piece of art cannot be adequately described using words. As a result, it developed an encoding module that translates a style image into the text domain using a CLIP image encoder. On the other hand, PHDDiffusion[26] enhances SD with a lightweight adaptive encoder, with the goal of extracting the necessary condition information (such as background style and image content) from the composite image. Nevertheless, previous diffusion-based techniques are unable to offer strong style guidance and maintain sufficient content in painterly image harmonization.

## 3 METHOD

Given an input image, denoted as $\mathbf{x}_G$, which can be either provided by the user or generated by the T2I-DM model using a semantic prompt $d$, our objective is to blend this image with another user-provided image, $\mathbf{x}_I$, using a binary mask $m$. The goal is to create a fused image, $\mathbf{x}_F$, where the content in the masked region, $\mathbf{x}_F \circ m$, closely resembles the structure and texture of the original image $\mathbf{x}_I$. In other words, we want $\mathbf{x}_F \circ m$ to be approximate to $\mathbf{x}_I$ (denoted as element-wise multiplication $\circ$). Additionally, the unmasked region should remain unchanged, meaning $\mathbf{x}_F \circ (1 - m) = \mathbf{x}_G$. It is also important to ensure that the blended region $\mathbf{x}_F \circ m$ and unchangeable region $\mathbf{x}_F \circ (1 - m)$ have a consistent style, resulting in a seamless and natural transition between the two regions.

To achieve this, we propose a method that utilizes the intermediate results from T2I-DM to guide the style transfer of the user-provided image. This is done by introducing a blending loss that consists of content loss, style loss, and stability loss.

### 3.1 Overall Workflow

As depicted in Figure 2, we leverage a pre-train T2I-DM architecture for our task. We have found that the VAE and Unet module in T2I-DM serve as excellent feature extractors. Prior to compositing the foreground and background images, we utilize the VAE image encoder to convert $\mathbf{x}_G$ and $\mathbf{x}_I$ into latent features $\hat{\mathbf{x}}_G$ and $\hat{\mathbf{x}}_I$. Then we conduct the noise augmentation to the output VAE feature based on the forward diffusion process. The aim of noise augmentation is to weaken the original style of the foreground image. But we should not inject too much noise during this process considering the controllability and time. Subsequently, we initialize a learnable latent feature $\hat{\mathbf{x}}_L$ with the same value as $\hat{\mathbf{x}}_I$ as a starting point. The objective of our method is to optimize the learnable feature $\hat{\mathbf{x}}_L$ so that it

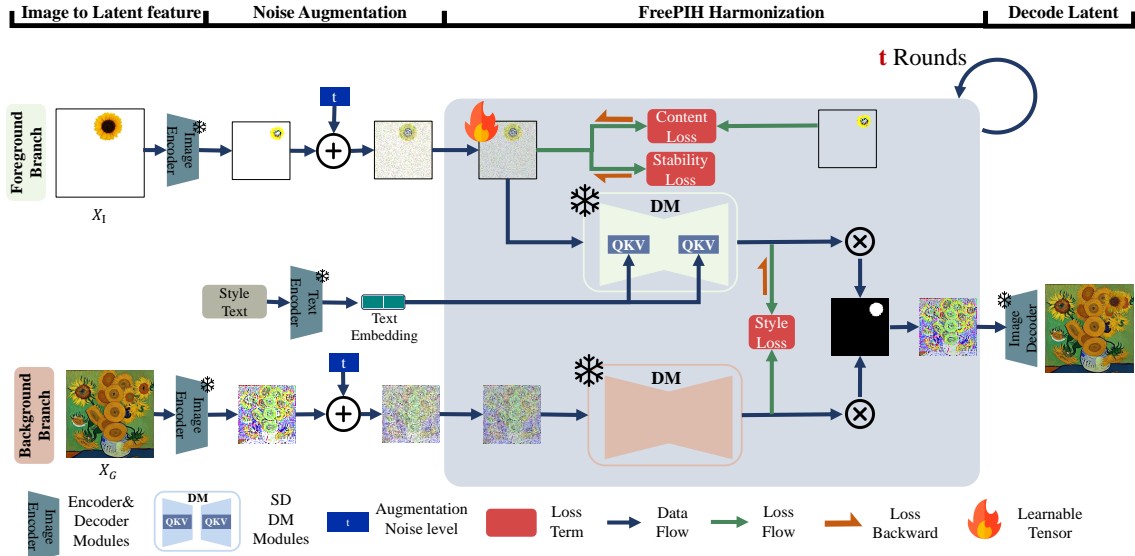

**Figure 2: The architecture of our FreePIH, we modify the pre-train stable diffusion model and add several loss terms to control the style transformation of foreground items.**

seamlessly integrates with $\hat{x}_G$ while preserving the majority of the features observed in $\hat{x}_I$. To achieve this goal, we introduce noise into the latent features $\hat{x}_G$ and $\hat{x}_L$. These augmented latent features are then fed through the DM network, and the resulting output is used to calculate the style loss. By performing backpropagation with both content loss and stability loss, we are able to update the learnable feature $\hat{x}_L$. We denote the updated result as $\hat{x}_L^{t-1}$. Next, we update $\hat{x}_G$ using the input mask, which can be expressed as follows:

$$\hat{x}_G^{t-1} = DM_\theta(\hat{x}_G^t) \circ (1-m) + DM_\theta(\hat{x}_L^{t-1}) \circ m, \quad (1)$$

where the $\hat{x}_L^{t-1}, \hat{x}_G^{t-1}$ then work as the input for next iteration. We repeat this process until $t = 0$. Finally, we have $\hat{x}_F = \hat{x}_G^0$, and we can use the VAE image decoder to decode the latent feature $\hat{x}_F$ to $x_F$, which is our final output. Note that in the compositing process, only the learnable parts are updated, while the DM, and VAE modules are frozen. Additionally, the compositing process can be completed within a few seconds, whereas fine-tuning a T2I-DM with Dreambooth may require as long as a day. This streamlined approach allows for efficient and timely image compositing while minimizing the overall computational burden.

## 3.2 Augmentation and Denoising

To capture the stylistic information with T2I-DM, we first conduct the augmentation over the input latent feature $\hat{x}_L, \hat{x}_G$ by injecting noise into the input latent feature. The results distribution after noise augmentation is as follows:

$$q(\hat{x}_L^t | \hat{x}_L) = \mathcal{N}(\hat{x}_L^t; \sqrt{\bar{\alpha}_t}\hat{x}_L, (1-\bar{\alpha}_t)\mathbf{I}), \quad (2)$$

where $t$ is a hyperparameter that controls the noise inject level. Typically, we set $t$ to be less than 0.2 times the total denoising steps ($T$), as depicted in Figure 3. This decision is based on three **key**

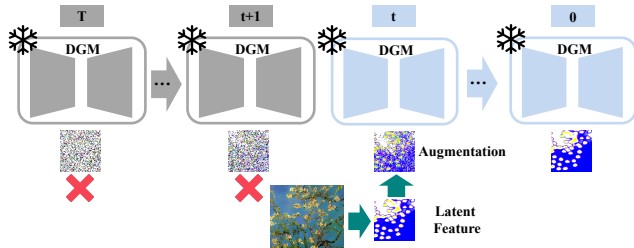

**Figure 3: Conduct the noise augmentation only on the last steps of the denoising process. We avoid the denoising calculation from $T$ to $t + 1$.**

**observations**. Firstly, previous studies [8, 14] have demonstrated that the denoising process in DM can be divided into two parts. After generating an image from a purely noisy image $\mathbf{x}^T$, the majority of the content remains unchanged from the first part. Subsequently, in the remaining denoising steps, the details and style are gradually refined. Therefore, to capture the style features, we only need to introduce a small amount of noise to modify the style. Secondly, our observations indicate that injecting excessive noise can disrupt the structure of the original image. Since the denoising process involves inherent randomness, injecting too much noise can lead to a loss of control over the image content and result in a different background. Thirdly, by ensuring that $t$ is less than $T$, we can expedite the inference process. If we inject $t$ steps of noise, we only need to denoise the same number of steps, thereby accelerating the overall process.

The denoising steps follow the normal DM process which we have:

$$q(\mathbf{x}^{t-1}|\mathbf{x}^t, c) = \mathcal{N}(\mathbf{x}^{t-1}|\boldsymbol{\mu}_\theta, \sigma^2\mathbf{I}),$$

$$\boldsymbol{\mu}_\theta = \frac{1}{\sqrt{\alpha_t}}\left(\mathbf{x}^t - \frac{1-\alpha_t}{\sqrt{1-\bar{\alpha}_t}}\boldsymbol{\epsilon}_\theta(\mathbf{x}^t, t, c)\right), \quad (3)$$

$$\sigma^2 = \frac{1-\bar{\alpha}_{t-1}}{1-\bar{\alpha}_t} \cdot \beta_t,$$

where $\boldsymbol{\epsilon}_\theta$ represents a neural network that takes the noised image $\mathbf{x}^t$, the time index $t$, and additional conditions as inputs, and predicts the noise that should be removed from $\mathbf{x}^t$. The solver used to sample $\mathbf{x}^{t-1}$ in Eq. ((3)) can be any solver proposed by previous works, such as DDPM[16], DDIM[41], DPM++[25], and so on.

Since the output of DM is already the noise version of the input (i.e., $\mathbf{x}^{t-1} = DM(\mathbf{x}^t, t)$), we can directly update $\hat{\mathbf{x}}_G$ using Eq. ((1)), which is shown as the intermediate line in Figure 2. The updated value can then serve as the input for the next iteration. As for $\hat{\mathbf{x}}_L$, we retain the noise injection steps with different $t$.

## 3.3 Loss

The fundamental concept behind style transfer is to utilize multiple loss functions to achieve a balance between various objectives, including transferring the visual style from image $\mathbf{x}_I$ to the style of image $\mathbf{x}_G$, preserving the structure and details of the input image $\mathbf{x}_I$, and seamlessly merging $\mathbf{x}_I$ into $\mathbf{x}_G$. The overall loss function can be represented as follows:

$$\mathcal{L} = \omega_{sty}\mathcal{L}_{sty} + \omega_c\mathcal{L}_c + \omega_{sta}\mathcal{L}_{sta}. \quad (4)$$

The loss functions for the mentioned objectives, denoted as $\mathcal{L}_{sty}, \mathcal{L}_c, \mathcal{L}_{sta}$, are used to measure and optimize the model's performance. These objectives represent different aspects and are assigned weights represented by $\omega_{sty}, \omega_c, \omega_{sta}$ to ensure a balanced combination of the losses. The distinguishing factor among various works lies in their formulation of these loss functions and the feature extractors employed. In our research, we take a different approach from previous studies that utilize VGG-Net as the feature extractor. Instead, we capitalize on the modules present in the T2I-DM, which themselves serve as exceptional multi-scale feature extractors. This allows us to harness the features extracted by these modules for the calculation of different loss functions, thus enhancing the overall performance of the model.

### 3.3.1 Style Loss.
To obtain the style feature of the background, denoted as $\hat{\mathbf{x}}_G$, we subject it to augmentation and input it into the DM. The output serves as the style feature representation of the background. To better utilize the multi-modal features of T2I-DM, we go a step further for the foreground. We incorporate textual information to provide DM with knowledge about the foreground item. This is achieved by encoding the text $c$ using the CLIP text encoder. The encoded text then guides the denoising process through the cross-attention mechanism.

$$Attn = Softmax\left(\frac{(\mathbf{W}_Q E_{img})(\mathbf{W}_K E_{txt})^T}{\sqrt{d}}\right)\mathbf{W}_V E_{txt}, \quad (5)$$

where $E_{img}$ is the intermediate feature of DM and $E_{txt}$ is the embedding result of CLIP text encoder. $\mathbf{W}_Q, \mathbf{W}_K, \mathbf{W}_V$ are attention weights. Finally, we calculate the style loss as:

$$\mathcal{L}_{sty} = ||G(DM_\theta(\hat{\mathbf{x}}_L^t, t, c)) - G(DM_\theta(\hat{\mathbf{x}}_G^t, t, c))||^2, \quad (6)$$

where $G(\cdot) = DM_\theta(\cdot)DM_\theta(\cdot)^T \in \mathbf{R}^{N\times N}$ is the Gram matrix. The advantage of using the Gram matrix is that it can remove the location impact on the style representation. Meanwhile, the product of DM feature and its transposition can turn the local statistics feature into a global feature[27].

### 3.3.2 Content Loss.
Since the latent feature $\hat{\mathbf{x}}_L$ serves as a comprehensive multi-scale content feature representation, we can calculate the content loss by measuring the difference between $\hat{\mathbf{x}}_L$ and $\hat{\mathbf{x}}_I$ as follows:

$$\mathcal{L}_c = ||\hat{\mathbf{x}}_L - \hat{\mathbf{x}}_I||. \quad (7)$$

By minimizing this loss term, we can ensure the content in the corresponding position of the target output have the close structure and detail as to the user providing $\mathbf{x}_I$.

### 3.3.3 Stability Loss.
To increase the stability of the output and reduce the ambiguity during the generation process, we add histogram loss[46] and total variation loss[19] into $\mathcal{L}_{sta}$. Histogram loss is calculated by:

$$\mathcal{L}_{his} = ||\hat{\mathbf{x}}_L - \mathbf{R}(\hat{\mathbf{x}}_L)||^2, \quad (8)$$

where $\mathbf{R}(\hat{\mathbf{x}}_L) = histmatch(\hat{\mathbf{x}}_L, \hat{\mathbf{x}}_G)$ is the histogram-remapped feature map by match $\hat{\mathbf{x}}_L$ to $\hat{\mathbf{x}}_G$.

Total variation loss is calculated by:

$$\mathcal{L}_{tv} = \sum_{i,j}(\hat{\mathbf{x}}_L(i,j) - \hat{\mathbf{x}}_L(i, j-1))^2 + (\hat{\mathbf{x}}_L(i,j) - \hat{\mathbf{x}}_L(i-1, j))^2, \quad (9)$$

where $\hat{\mathbf{x}}_L(i,j)$ represent the feature in the position $(i, j)$.

Finally, we have:

$$\mathcal{L}_{sta} = \lambda_{his}\mathcal{L}_{his} + \lambda_{tv}\mathcal{L}_{tv}, \quad (10)$$

where $\lambda_{his}, \lambda_{tv}$ are two hyperparameters to balance the influence of these two loss terms. $\mathcal{L}_{sta}$ term can improve the compositing result by producing smoother output.

## 3.4 Optimization

For the optimization process, we have carefully selected the values of the weighting parameters: $\omega_{sty}$ is set to $1e7$, $\omega_c$ is set to $1e1$, and $\omega_{sta}$ is set to 1. Additionally, we perform 5 rounds of optimization in each iteration, and instead of using the Adam solver, we utilize a quasi-Newton solver called L-BFGS to minimize the loss function instead of Adam solver as we found failed to composite the foreground image into the background with the same number of optimization rounds as the L-BFGS solver.

## 3.5 Second Stage Refinement.

As shown Figure 4, though we can adapt the style of the foreground image to that of the background image with our loss terms, there are still some distortions in the output images. In order to enhance the quality of the fusion image and minimize artifacts in the transition areas between the foreground and background images, we employ a square mask that encompasses the entirety of the foreground region along with a portion of the background region. Subsequently, we introduce a slight amount of noise into this region and employ the same T2I-DM to denoise it. During this stage, we extend the mask to a square region as some distortions may be outside the original mask region. Meanwhile, we eliminate all loss terms and solely

**First Stage** **Second Stage**

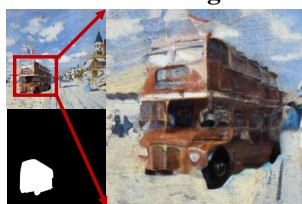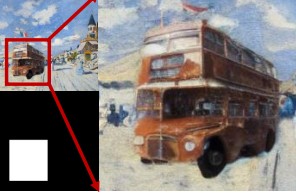

**Figure 4: Second stage refinement are adopted to enhance the quality of the fusion image.**

---

**Algorithm 1** FreePIH

**Input**: Foreground $\mathbf{x}_I$, Background $\mathbf{x}_G$, Mask $m$, Style text $c$,
        Pre-train $T2I - DM_\theta$, Augmentation strength $t$
**Output**: Fused image $\mathbf{x}_F$

1: Let ENC be the VAE encoder of $T2I - DM_\theta$.
2: Let DEC be the VAE decoder of $T2I - DM_\theta$.
3: Let $DM$ be the Unet of $T2I - DM_\theta$.
4: $\hat{\mathbf{x}}_G = ENC(\mathbf{x}_G), \hat{\mathbf{x}}_I = ENC(\mathbf{x}_I)$.
5: $\hat{\mathbf{x}}_L = \hat{\mathbf{x}}_I$.
6: $Optimizer = LBFGS(\hat{\mathbf{x}}_L)$
7: **for** $i \in [t, ..., 0]$ **do**
8:     $\hat{\mathbf{x}}_L^t = Augmentation(\hat{\mathbf{x}}_L, t)$
9:     $\hat{\mathbf{x}}_G^t = Augmentation(\hat{\mathbf{x}}_G, t)$
10:     $\mathcal{L}_{sty} = StyleLoss(\hat{\mathbf{x}}_L^{i-1}, \hat{\mathbf{x}}_G^{i-1})$
11:     $\mathcal{L}_c = MSELoss(\hat{\mathbf{x}}_L, \hat{\mathbf{x}}_G)$
12:     $\mathcal{L}_{sta} = StabilityLoss(\hat{\mathbf{x}}_L)$
13:     $\mathcal{L} = \omega_{sty}\mathcal{L}_{sty} + \omega_c\mathcal{L}_c + \omega_{sta}\mathcal{L}_{sta}$
14:     $Backward(\mathcal{L})$
15:     $Step(Optimizer)$
16:     $\hat{\mathbf{x}}_G = DM(\hat{\mathbf{x}}_G^i, i, c) \circ (1 - m) + DM(\hat{\mathbf{x}}_L^i, i, c) \circ m$
17: **end for**
18: $\hat{\mathbf{x}}_F = \hat{\mathbf{x}}_G$
19: $\mathbf{x}_F = DEC(\hat{\mathbf{x}}_F)$
20: **return** $\mathbf{x}_F$.

---

retain the text prompt, making the denoising process akin to the SDEdit process.

## 4 EXPERIMENT

The foreground items in our experiment are extracted from COCO datasets[22], which is a large-scale object detection, segmentation, and captioning dataset. COCO has 80 object categories, 1.5 million object instances, we use the pycocotools library to fetch content and mask from this dataset. The background images are randomly selected from LAION[39] (acronym for Large-scale Artificial Intelligence Open Network) which is a number of large datasets of image-caption pairs. Now the dataset contain more than 5 billion image-text pairs of various artistic styles.

### 4.1 Baselines

For comparison, we choose several baseline methods including non-DM-based Poisson image editing (PIE)[33], Deep Image Blending

(DIB)[49], PHDNet[3] and DM-based SDEdit[29], SD-Text[37], and BlendDM[1]. Meanwhile, following the previous works, we also use recent work CDC[12], InST[50], PHDiff[26] as the baselines.

Among these, SD-Text necessitates detailed textual information about output images, while other DM-based methods only require simple textual input about the foreground items. All of the codes and pre-trained weights have been made available by the authors. For all the baselines, we obtained the code from their official GitHub repositories. For methods PIE and DIB, they solely utilize the pre-trained VGG-19. In the case of PHDNet, it requires VGG-19 and another specialized neural network trained on large image datasets. We acquired their pre-trained model from their repositories to conduct the evaluation. As for SDEdit, it utilizes the pre-trained stable diffusion model with version sd-v1-5. In contrast, for SD-text, we need to conduct several fine-tuning iterations of the pre-trained stable diffusion model with various foreground items, akin to the Dreambooth setting[38]. In the context of text-driven image editing BlendDM, we obtained their pre-trained unconditional diffusion model to carry out the subsequent evaluation. CDC, InST, and PHD-iff are all grounded on stable diffusion 1.4. Notably, for InST, we acquired the andre-derain embedding provided by the authors for our evaluation. For PHDiff, we utilized SD1.4 and the PHDiffusion-WithRes checkpoints provided in the GitHub repository.

For our FreePIH, we use the pretrain model sd-v1-5 released by Stability AI, all the parameters of every SD modules are frozen during our inference and we only update the latent feature of foreground content so we are actually training-free. Specially, in our modified version of the SD, we have introduced several additional loss terms to optimize the latent feature, with the goal of transferring its style. It is worth noting that the only learnable part in our pipeline is the latent feature of the foreground items. During the inference process, we freeze all the modules including the Image Encoder, Text Encoder, DGM, and Image Decoder. This approach allows us to avoid heavy training costs and enables quick utilization for painterly image harmonization sourced from the internet. We implement FreePIH and test all the baselines on ubuntu 18.04 LTS operation system, with 64GB memory, a 12900K Intel CPU @3.20GHz and an NVIDIA RTX 4090 GPU. The pytorch version is 2.0.0. And the output image size is 512 × 512.

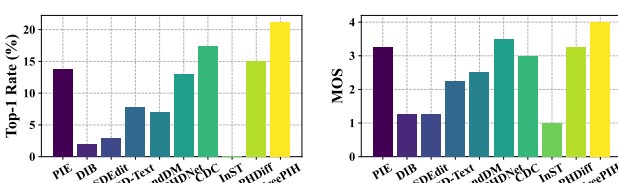

**Figure 5:** *Left* **The average Top-1 rate of all the baselines.** *Right* **The mean opinion score of different baselines, with a score from one** *Bad* **to five** *Excellent*.

### 4.2 Quantitative Evaluation

As mentioned in prior research, accurately computing common metrics such as MSE, PSNR, and SSIM for harmonized images proves to be challenging, particularly when foreground and background

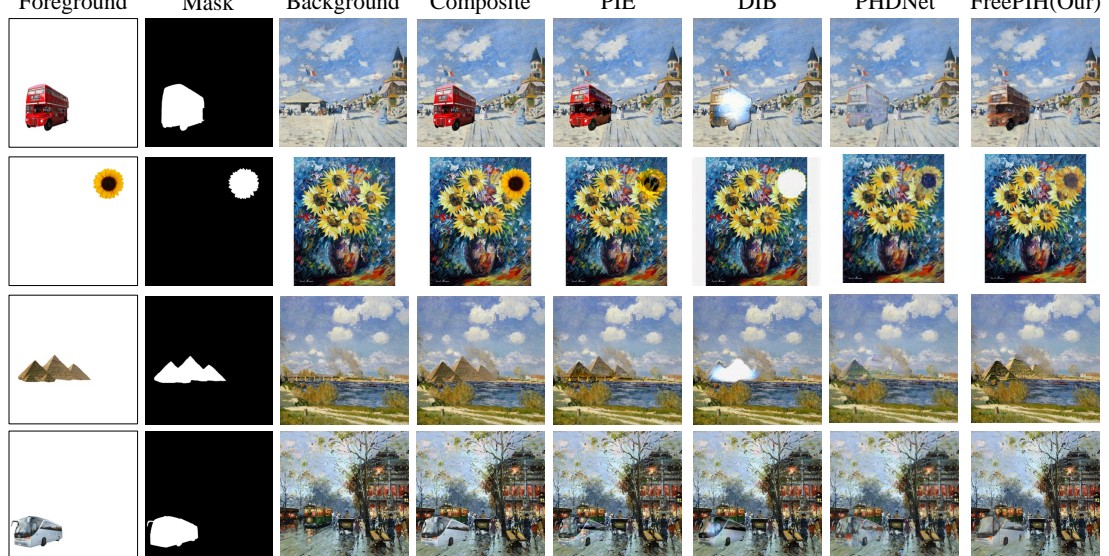

**Figure 6: Example results of non-DM-based painterly image harmonization baselines and our FreePIH.**

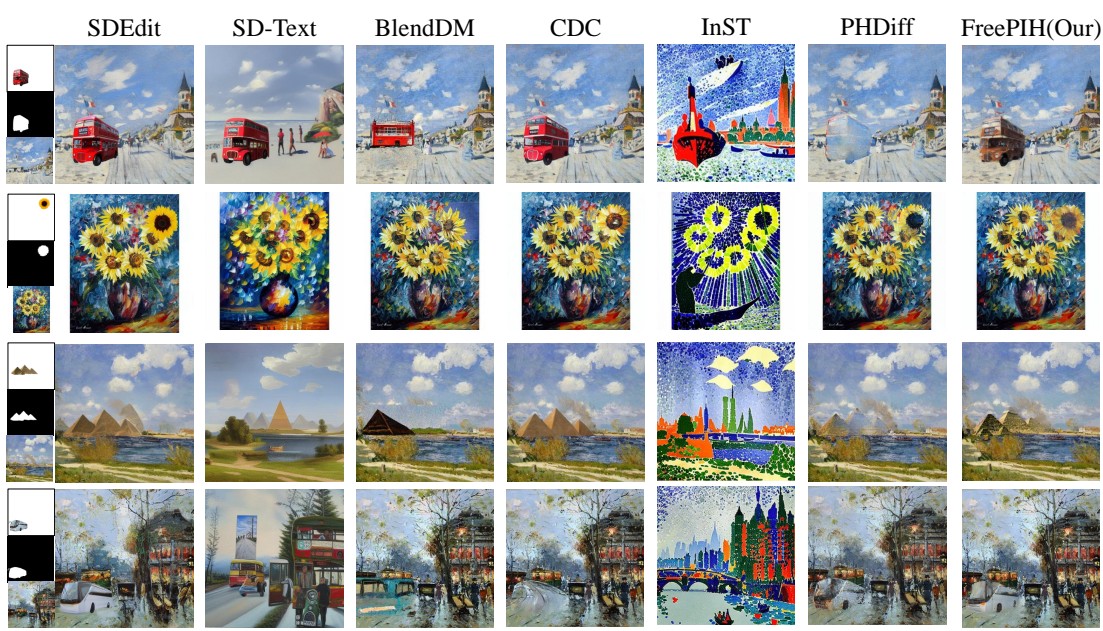

**Figure 7: Example results of DM-based painterly image harmonization baselines and our FreePIH.**

images are randomly selected. To surmount this challenge and quantitatively assess performance, we conducted a user study. Initially, we randomly selected 100 foreground and background pairs from the COCO2017 evaluation dataset [22] and LAION 5B dataset[39], respectively. Subsequently, we applied our method, alongside all the baselines, to generate the composited images. Using a subset of these images, we designed a questionnaire in which each query featured the same images but produced by different methods. Participants were tasked with selecting the top-1 harmonious image,

and a subset of the respondents also rated the images on a scale from one (poor) to five (excellent).

In total, we amassed 365 votes from 73 users. Subsequently, we aggregated the average top-1 rate and computed the Mean Opinion Score (MOS) based on the users' ratings[28]. The outcomes are illustrated in Figure 5, where we obtained approximately a 20% top-1 Rate across all the tests, while the second-best baseline CDC received about a 17% top-1 Rate. Furthermore, based on the voting results, our FreePIH garnered the highest MOS score (4), surpassing

other baselines such as CDC (3.5), PHDiff (3.2), and so forth. The voting outcomes underscore that our FreePIH method outperforms the other baselines in our human evaluation test.

**Table 1: Important factor impact the top-1 decision**

| Visual factors | Rank |
|---|---|
| Similar artistic effect on fore/back-grounds image style | 1 |
| Same color tone and saturability of fore/back-grounds image | 2 |
| No obvious error at the connect region | 3 |
| Foreground image maintain the original details | 4 |
| Background image maintain the original details | 5 |

Furthermore, we conducted a survey of 30 participants on their preferences of the most important factor when we pick the top-1 harmonization images and rank them according to the vote results. The results are shown in Table 1. The results show that artistic effect, the color tone and saturability siginificantly impact the participants' preference when they conduct the user study.

### 4.3 Qualitative Evaluation

The qualitative assessment of non-diffusion-based baselines is depicted in Figure 6. In our experiment, we observed that PIE retains the original foreground feature after optimization; however, the optimized foreground occasionally becomes transparent. As a result, the foreground content fails to obscure the background image, leading to the visibility of background people, which appears unnatural. Furthermore, in some instances, DIB is unable to complete the transformation, as evidenced in the first three rows. Additionally, certain foreground regions are heavily influenced by the background style, resulting in a clear separation between different areas. For example, in row 4, the center of the bus exhibits discordant blue and appears brighter than the rest. While PHDNet demonstrates improved performance compared to the previous baselines, there are instances where the foreground becomes excessively harmonious, leading to the loss of its original texture and details. This is evident in the first and third rows, where the bus and pyramid assume the color of the sky. Particularly in row 3, the other two pyramids blend into the sky, leaving only one pyramid noticeable at first glance.

In the Figure 7, SDEdit effectively preserves the texture of the foreground items but falls short in transforming their style. Conversely, with SD-Text the foreground and background may significantly deviate from our expectations. BlendDM can introduce a new item into the designated mask region, but the generated item may not align with the one originally inputted. CDC achieves the most harmonious results in the second image. However, the harmonization results in other cases are not satisfactory. InST can result in a lack of control over the entire images, leading to entirely different foreground/background images. PHDiff encounters the same issue as PHDNet, where the foreground becomes excessively transparent, leading to the loss of original details.

In contrast, our FreePIH effectively preserves the texture and structure of both the foreground and background, seamlessly fusing them. For instance, FreePIH transforms the original deep red to a red similar to the house in row 1 and successfully optimizes the boundary while preserving the colorful foreground items in rows 2-4.

### 4.4 Ablation Study

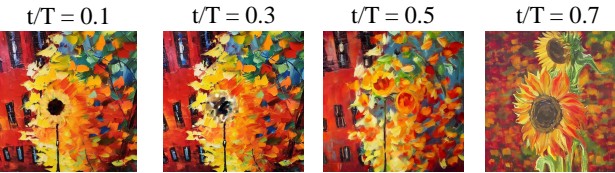

| t/T = 0.1 | t/T = 0.3 | t/T = 0.5 | t/T = 0.7 |
|---|---|---|---|

**Figure 8: Ablation results of noise step.**

To evaluate the impact of certain hyperparameters, we conducted ablation experiments on the noise level. The results of these experiments are shown in Figure 8. We adjusted the noise level (represented by t/T) for the same harmonization task. We added different levels of loss in the forward process. As depicted in the figure, when the noise level is small, we were able to successfully fuse the foreground sunflower with the background painting. However, as the noise level increased, the texture and details of the foreground sunflower gradually deteriorated. Ultimately, when t/T reached values less than or equal to 0.5, the foreground became unrecognizable sunflowers.

In particular, we also conduct an ablation study to analyse the impact of different loss term to the final output. As illustrated in Figure 9, using only content loss preserves the details of the foreground, but the result appears inharmonious. If we only use content loss, the details of foreground image can be perversed, however the composition results seems to be inharmonious. If we only use style loss, the details of foreground may lost finally. As for the stability loss, it seems our methods may not be sensitive to this loss. But empirically, the weight of this loss should not be too large.

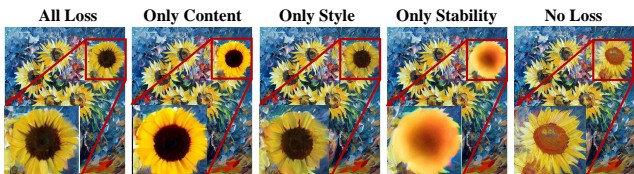

| All Loss | Only Content | Only Style | Only Stability | No Loss |
|---|---|---|---|---|

**Figure 9: Ablation of different loss.**

## 5 CONCLUSION

In this paper, we present a pioneering approach to painterly image harmonization using diffusion-based techniques without the need for training. Our method leverages the observation that the final stages of the diffusion generation process capture crucial stylistic information in images. By utilizing the output features of the Diffusion Model (DM), we achieve a seamless transformation of foreground styles into background styles, resulting in harmonious image compositions. Notably, our method stands out from other baselines, particularly DM-based approaches, as it eliminates the requirement for extensive fine-tuning or training auxiliary modules on new data. It can be conveniently employed as a plug-in module for existing stable diffusion frameworks. Through both quantitative and qualitative evaluations, we demonstrate the superiority of our proposed FreePIH.

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
