# OpenReview forum: "FreePIH: Training-Free Painterly Image Harmonization with Diffusion Model"
_acmmm.org/ACMMM/2024/Conference — MM2024 Oral_

### Official Review · Reviewer_9SA1 · 2024-04-27

**Rating:** 3
**Confidence:** 3

**Summary:**

This paper provides an efficient training-free painterly image harmonization (PIH) method, dubbed FreePIH, that leverages only a
pre-trained diffusion model to achieve state-of-the-art harmonization result. The auther conduct noise augmentation on the latent features and leverage th ecorresponding output to accurately capture stylistic information.

**Strengths:**

This paper is well writed and easy to follow.
The qualitative and quantitative experiment seem not bad.

**Limitations:**

1.In Fig. 7, the author provides some comparison experiments with PHDiff, but they are not competitive in the effects of partial graphs, such as the fourth row. Please consider providing some more representative results to demonstrate the validity of the authors' proposed method.
2.The method proposed in this paper lacks some innovation.
3.This paper is more like an engineering application.
4.If possible, the author can provide an online system for testing.

**Suitability:**

2

---

### Official Review · Reviewer_mZUP · 2024-05-17

**Rating:** 5
**Confidence:** 3

**Summary:**

The paper introduces FreePIH, an innovative training-free method for painterly image harmonization. It utilizes a pre-trained diffusion model to achieve state-of-the-art harmonization results without needing auxiliary networks or fine-tuning large pre-trained backbones. FreePIH leverages the denoising process as a plug-in module for transferring the style of the foreground image to match a painterly-style background. The key insight is that the final steps of the denoising process are crucial for stylistic information. The method augments the latent features of both foreground and background images with Gaussians, enabling direct denoising-based harmonization. To ensure the fidelity of the harmonized image, multi-scale features are employed to maintain content consistency and foreground object stability in the latent space while aligning the styles of both foreground and background. Additionally, text prompts are integrated to enhance the structural and textural details, further improving the generation quality.

**Strengths:**

1.The proposed FreePIH method is a novel training-free approach for painterly image harmonization. It stands out by utilizing a pre-trained diffusion model, avoiding the need for training auxiliary networks or fine-tuning large pre-trained backbones, which is a significant advancement over existing methods.

2.The paper clearly explains the method and its implementation. The description of leveraging denoising, augmenting latent features, and integrating text prompts for improved generation quality is well-articulated, making the contributions and innovations of the work easy to understand.

3.FreePIH has practical applications in image editing, graphic design, and any scenario where harmonizing foreground objects with painterly backgrounds is required. Its training-free nature makes it accessible and easy to integrate into various image processing workflows.

**Limitations:**

1. Can you explain more about introducing a small amount of noise to capture the style features?

2. More image notations such as $\hat{X}_G$, $\hat{X}_I$ should be presented in Figure 2 to clear understand the work flow.

3. I'm not clear about the multi-scale features in the abstract, which are mot mentioned a lot in the main text.

**Suitability:**

2

---

### Official Review · Reviewer_1ECF · 2024-05-24

**Rating:** 4
**Confidence:** 3

**Summary:**

The paper introduces FreePIH, a training-free painterly image harmonization method that utilizes a pre-trained diffusion model. This approach eliminates the need for training auxiliary networks or fine-tuning large pre-trained models typically required for harmonizing foreground objects with painterly-style background images. The authors discovered that the final steps of the denoising process are highly correlated with stylistic information, leading to a method that augments latent features of both foreground and background with Gaussians for direct harmonization via denoising. To ensure image fidelity, multi-scale features are employed, and text prompts are integrated to enhance structural and textural details in generation. Evaluations on COCO and LAION 5B datasets demonstrate the method's superiority over existing baselines.

**Strengths:**

(1). Empirical validation: The paper provides empirical evidence of its method's effectiveness through evaluations on established datasets like COCO and LAION 5B. This validation is a strength, as it demonstrates the method's real-world applicability and performance. (2).Training-free approach: The training-free nature of the method is a significant advantage, as it simplifies the application of the technique and makes it more accessible.

**Limitations:**

(1).The paper's approach of introducing noise to an image at a certain timestep (as shown in Figure 3) is not innovative, as this is a common operation found in publicly available codebases, typically applied as a noise strength parameter. Presenting this as a significant contribution may not hold up to scrutiny. Besides, The term "augmentation" in Section 3.2 is quite confusing. My understanding is that what the authors actually intend to describe is simply the forward noise addition process in the diffusion, just adding noise to some intermediate timestep. (2). Efficiency of the optimization method: The paper employs an optimization method to update a learnable tensor, yet it does not elaborate on the inference time compared to other existing techniques. (3).The paper did not conduct an ablation study to examine the effect of incorporating the style text as guidance.

**Suitability:**

3

---

### Official Review · Reviewer_GqYr · 2024-05-25

**Rating:** 4
**Confidence:** 3

**Summary:**

This paper proposes a new image harmonization method called Free Painterly Image Harmonizationan (FreePIH)  for training-free painterly image harmonization. The key idea is to leverage a pre-trained diffusion model to achieve state-of-the-art harmonization results without the need for training additional networks or fine-tuning a large pre-trained backbone.

The approach is designed to work as a plug-in module, enabling  image harmonization without the need for collecting new data, training auxiliary networks, and fine-tuning pre-trained models.

**Strengths:**

The paper provides some implications on difussion model, finding that the last few steps of the denoising (generation) process in diffusion models strongly correlate with the stylistic information of images. Based on it, the paper proposes to augment the latent features of both the foreground and background images with Gaussians to achieve direct denoising-based harmonization. This approach is novel and has the potential to effectively blend foreground objects into painterly-style background images.

To ensure the fidelity of the harmonized image, the paper utilizes multi-scale features to enforce the consistency of content and stability of foreground objects in the latent space. The  approach helps maintain the structural and textural details of the foreground objects while harmonizing them with the background.

Experimental comparisons are given to demonstrate that the proposed FreePIH method achieves better performance than existing  representative baselines.

**Limitations:**

The paper focuses narrowly on a specific aspect of the research area, which may limit its applicability to broader contexts. More discussion of potential extensions and generalizations would strengthen the work.

The paper lacks of theoretical analysis on its main contribution that why the proposed model can work as a plug-in module, while without degnerating performance on image harmonization.

While the paper discusses potential limitations and extensions, it lacks a clear roadmap for future research.  No new dataset is delevered in the paper.

**Suitability:**

3

---

### Official Review · Reviewer_z87v · 2024-05-26

**Rating:** 4
**Confidence:** 3

**Summary:**

This paper provides an efficient training-free painterly image har monization  that leverages only a pre-trained diffusion model to achieve state-of-the-art harmonization results. Unlike previous methods that require either training auxiliary networks or fine-tuning a large pre-trained backbone,or both, to harmonize a foreground object with a painterly-style background image, the proposed tames the denoising process as a plug-in module for foreground image style transfer. Quantitative and qualitative evaluations on COCO and LAION 5B datasets demonstrate the effectiveness of the proposed method.

**Strengths:**

1. The method seems simple yet effective.
2. User study shows better visual quality of the proposed method.
3. Using a pre-trained diffusion model produces better performance.

**Limitations:**

1. Since diffusion-based method have widely used in this field, what are the differences? I cannot find more insight about it.
2. Why not test on other fidelity metrics, like FID or LIPS.
3. There exist plenty of style methods. Why no compare with other mainstream image style methods. What is the effects of style method used.

**Suitability:**

2

---

### Meta-Review · Area_Chair_qQcB · 2024-06-26

**Recommendation:** Accept (Oral)
**Confidence:** 5

**Metareview:**

Most reviewers vote to accept this paper. This paper proposes a novel training-free painterly harmonization method, which avoids the burden of training or finetuning. The idea is simple and effective. The harmonized results are visually appealing.  I agree with most reviewers to accept this paper.